# Involvement of Diamine Oxidase in Modification of Plasma Membrane Proton Pump Activity in *Cucumis sativus* L. Seedlings under Cadmium Stress

**DOI:** 10.3390/ijms24010262

**Published:** 2022-12-23

**Authors:** Małgorzata Janicka, Małgorzata Reda, Natalia Napieraj, Adrianna Michalak, Dagmara Jakubowska, Katarzyna Kabała

**Affiliations:** Department of Plant Molecular Physiology, Faculty of Biological Sciences, University of Wrocław, Kanonia 6/8, 50-328 Wrocław, Poland

**Keywords:** DAO, Cd, PM H^+^-ATPase, cucumber, heavy metals, hydrogen peroxide, nitric oxide

## Abstract

Cucumber (*Cucumis sativus* L.) is a crop plant being the third most-produced vegetable developed as a new model plant. Heavy metal pollution is a serious global problem that affects crop production. An industrial activity has led to high emissions of Cd into the environment. Plants realize adaptive strategies to diminish the toxic effects of Cd. They can remove excess toxic ions of heavy metals from the cytoplasm to the outside of cells using the metal/proton antiport. The proton gradient needed for the action of the antiporter is generated by the plasma membrane (PM) H^+^-ATPase (EC 3.6.3.14). We have shown that treatment of cucumber plants with Cd stimulated the diamine oxidase (DAO, EC 1.4.3.6) activity in roots. Under cadmium stress, the PM H^+^-ATPase activity also increased in cucumber seedlings. The stimulating effect of Cd on the PM H^+^-ATPase activity and expression of three genes encoding this enzyme (*CsHA2*, *CsHA4*, *CsHA8*) was reduced by aminoguanidine (AG, a DAO inhibitor). Moreover, we have observed that H_2_O_2_ produced by DAO promotes the formation of NO in the roots of seedlings. The results presented in this work showed that DAO may be an element of the signal transduction pathway, leading to enhanced PM H^+^-ATPase activity under cadmium stress.

## 1. Introduction

Abiotic stresses are considered as major limiting factors that affect crop production. As estimated, around 90% of all arable lands are subjected to environmental stresses such as drought, low or high temperature, salinity, heavy metal exposure, and others [1]. In recent years, heavy metal pollution has become a more serious global problem. Among the heavy metals, cadmium is one of the most phytotoxic agents. Cd is easily taken up by plant roots and can accumulate for long periods of time inside a food chain (in plants and animals) [2]. This toxic metal has a negative effect on the growth and development of plants [3]. Cd reduces seed germination, early seedling growth, and plant bio-mass [4]. It was shown that the toxic effect of Cd ions on photosynthesis plays a significant role in the inhibition of plant growth [5]. Cd has been found to mediate oxidative stress, but unlike other heavy metals, it does not directly affect the production of reactive oxygen species (ROS) through the reaction of Fenton and/or Haber Weiss. Nevertheless, the production of ROS such as hydroxyl radical, hydrogen peroxide, singlet oxygen, and superoxide radicals after Cd exposure has been reported in plants [6].

Plants have developed several adaptive strategies to fight the toxic effects of Cd. An important element of detoxification is the exclusion of Cd outside the cell or its accumulation in metabolically less active parts of the plant [4]. Removing Cd from the cell is related to immobilizing it in the cell wall or chelation. Cd could be chelated in the apoplast and this limits its harmful effects in the cell [7]. Plants can remove excess toxic heavy metal ions from the cytoplasm to the outside of cells using the heavy metal/proton antiport systems. The cation diffusion facilitator (CDF) family proteins are membrane divalent cation transporters that transfer metal ions out of the cytoplasm into extracellular space or into vacuoles, and they act as metal^2+^/H^+^ antiporters [8]. The mechanism of Cd detoxification that relies on Cd^2+^/H^+^ antiport activity in plant plasma membrane has been reported [9]. Plant CDFs are called metal tolerance proteins (MTPs). The proton gradient needed for the action of the MTP antiporter located in the plasma membrane is generated by the plasma membrane proton pump (PM H^+^-ATPase).

PM H^+^-ATPase is a functional single polypeptide chain with a mass of about 100 kDa. The protein can oligomerize to form dimeric and hexameric complexes [10]. PM H^+^-ATPase belongs to the P-type ATPase superfamily. Among the P-type ATPases identified in plants, none exchange sodium and potassium, as does the animal Na^+^/K^+^-ATPase. Plants possess proton vector pumping ATPase (H^+^-ATPase), which combines ATP hydrolysis with proton transport from the cytoplasm out of the cell to the apoplast, thus creating an electrochemical gradient across the plasma membrane [11,12]. Generation of an electrochemical gradient across membrane results in a proton-motive force that is used by secondary transport for assimilation of various nutrients, and on the other hand, for releasing ions and toxic substances from cells [13]. Aside from the regulation of growth and development processes, the PM H^+^-ATPase also plays a role in the plastic adaptation of plants to changing conditions, especially stressogenic ones.

The plant PM H^+^-ATPases are a multigene family of proteins, and a total of 10 genes have been found in *Cucumis sativus*, seven of which (*CsHA1*, *CsHA2*, *CsHA3*, *CsHA4*, *CsHA8*, *CsHA9*, *CsHA10*) are expressed in the roots [14]. The PM H^+^-ATPases are differentially expressed according to environmental factors. Several studies have indicated that the H^+^-ATPase gene expression might be changed by environmental factors: salt [15], low temperature [16], heavy metals [17,18], dehydration [19], or mechanical stress [20]. Aside from the genetic regulation of a proton pump, its activity might be fast modulated posttranslationally at the protein level, mainly through reversible phosphorylation. PM H^+^-ATPase contains ten transmembrane domains. The catalytic domain is between transmembrane domains 4 and 5. The N- and C-termini of the protein are located on the cytoplasmic side [11]. The C-terminal region is involved in enzyme regulation. Phosphorylation of the penultimate residue, a Thr, and the subsequent binding of regulatory 14–3–3 proteinresult in enzyme activation.

It has been reported that PM H^+^-ATPase activity is changed due to Cd action. The effect of metal on plasma membrane H^+^-ATPase activity depends on the time of plant exposure as well as heavy metal concentration. In short time (up to a few hours) Cd-exposed plants, the inhibition of PM H^+^-ATPase activity was observed [17,21]. However, a longer time of plant treatment with Cd led to increased activity of the enzyme [9,18]. A plasma membrane is the first cellular structure to be exposed to the toxic effects of cadmium. This metal often causes membrane damage, which increases the permeability of ions and the loss of valuable substances [21]. An increase in PM H^+^-ATPase activity is necessary to generate a proton gradient across the plasma membrane in order to both replenish lost substances and get rid of excess toxic ions.

Plants also adapt to Cd toxicity by activating signaling pathways that allow them to function (grow and develop) under cadmium stress conditions. ROS-signaling is very important in this process. It was previously mentioned that the accumulation of cadmium caused the enhanced production of ROS, which can act as signaling molecules in the plants’ defense [22]. There are many potential sources of hydrogen peroxide production in plant cells. One of them is the degradation of polyamines (PAs), which contributes to an increase in extracellular H_2_O_2_ level. PAs play a crucial role in the responses of plants to abiotic stresses. They are oxidatively deaminated by amine oxidases including flavin adenine dinucleotide (FAD)-dependent polyamine oxidases (PAO, EC 1.5.3.3) and copper amine oxidases (CuAO, EC 1.4.3.6), also called diamine oxidases (DAO). DAO and PAO are responsible for the oxidation of PAs in plants, which occurs with production of H_2_O_2_ [23]. Amine oxidases require oxygen and water to deaminate PAs, and forming H_2_O_2_ and aldehyde as a result of cutting off the amino group [24].

DAO is the enzyme that transforms a primary amino group –NH_2_ with copper as a cofactor. Oxygen is the acceptor of the charge carried by this enzyme, which shows a preference for putrescine as a diamine [25]. DAO converts putrescine to 4-aminobutanal, which spontaneously changes into a cyclic compound Δ1-pyrroline. The other products are ammonia and hydrogen peroxide [24,26,27]. This oxidase tends to form homodimers with the weight of each subunit about 70–90 kDa. There are 33 conserved amino acid residues and the 2,4,5-trihydroxyphenylalanine quinone cofactor in each subunit near the catalytic site. Additionally, a coordination bond between the copper (II) ion and three histidine residues is formed in each subunit [24]. DAO is located in the cell wall and loosely associated with it [24,27]. PAO catabolize higher polyamines (e.g., spermmidine and spermine). It was reported that PAO is highly expressed in monocots whereas DAO is present at high levels in dicotyledons [25], as shown in the examples of species from the *Fabaceae* family (e.g., peas (*Pisum*), chickpeas (*Cicer*), lentils (*Lens*), soybeans (*Glycine*)) [25]. Diamine oxidase is also often found in rapidly growing tissues [24].

The involvement of amine oxidases in polyamine catabolism contributes to the adaptation of plants to adverse environmental conditions. Under aluminum stress, an increase in diamine oxidase activity was observed in pea root nodules. This increase in DAO activity was responsible for enhanced accumulation of hydrogen peroxide in nodules [28].

Aside from H_2_O_2_, PAs produce nitric oxide (NO) during their catabolism. Tun et al. [29] reported that PAs induced NO biosynthesis in *Arabidopsis thaliana* seedlings. NO is a key signaling molecule in plants, regulating a lot of physiological processes. Moreover, NO plays an important role in the regulation of plant responses to both abiotic and biotic stress conditions. DAO could participate in nitric oxide production in plants under environmental stresses. CuAO8 regulates arginine-dependent NO generation in *Arabidopsis thaliana* [30]. It was shown that the *cuao8* mutant lines displayed a decreased NO production in seedlings after elicitor and salinity treatment. The review of Gill et al. [31] showed the importance of NO as a Cd stress modulator in crop plants.

In our earlier study, we showed that under cadmium stress, the PM H^+^-ATPase activity increased in the roots of cucumber seedlings [18]. It was found that signaling molecules NO and H_2_O_2_ are important in the modification of plasma membrane proton pump activity under abiotic stress conditions including salinity and low temperature [15]. Considering this, we decided to verify whether amine oxidases can contribute to the modification of the PM H^+^-ATPase activity in *Cucumis sativus* L. seedlings under cadmium stress. For this purpose, we performed experiments in which we determined the activities of DAO and PAO in cucumber roots; DAO activity in plants treated with Cd and/or AG (aminoguanidine, DAO inhibitor); activity of plasma membrane H^+^-ATPase and expression of genes encoding PM H^+^-ATPase in cucumber seedlings treated with Cd and/or AG; NO and H_2_O_2_ level in plants treated with Cd and/or AG. The obtained results suggest that DAO may be an important element of the signal transduction pathway, leading to an increase in PM H^+^-ATPase activity under cadmium stress.

## 2. Results

### 2.1. Activities of DAO and PAO in Cucumber Roots

There are two main types of amine oxidases (AOs) in plants including copper-containing (DAO) and FAD-dependent (PAO). In our study, activities of both AOs, DAO and PAO, were measured in the roots of cucumber seedlings grown under the control (non-stress) conditions. It was shown that DAO activity was almost sixteen times greater than PAO activity (Figure 1).

We observed that the treatment of plants with cadmium (10 µM Cd) changed the activity of DAO but had no effect on the activity of PAO in cucumber roots. PAO activity was always at a very low level, regardless of the plant treatment with cadmium (Repository 1). On the other hand, exposure of plants to cadmium stimulated twice DAO activity (Figure 2). To make sure that the determined activity, stimulated by Cd, is actually DAO activity, we used a DAO inhibitor (i.e., 0.1 mM aminoguanidine (AG)). The addition of AG greatly (about 90%) reduced DAO activity in both the control and cadmium-treated plants (Table 1).

### 2.2. Activity of Plasma Membrane H^+^-ATPase in Roots of Cucumber Seedlings Treated with Cd and/or AG

Treatment of the cucumber seedlings with cadmium increased the H^+^-ATPase activity in the plasma membrane fraction isolated from roots (Figure 3A,B). Cd activated the H^+^-pumping into vesicles to a much greater extent than the hydrolysis of ATP. The proton transport was stimulated by 89% whereas the hydrolytic activity by 37%. To verify whether the increased enzyme activity could be attributed to DAO action, the effect of aminoguanidine on the proton pump was studied in seedlings under the control and cadmium stress conditions. As we earlier indicated, treatment of the plants with cadmium and then transferring them to the control conditions increased the activity of the plasma membrane proton pump [18]. For this reason, AG was added to the medium after cadmium removal. It was found that the stimulating effect of cadmium on the plasma membrane H^+^-ATPase activity (both H^+^ transport and ATP hydrolysis) was totally abolished by AG (Cd/AG). In contrast, aminoguanidine had no effect on PM H^+^-ATPase activity in the control plants, not treated with Cd.

### 2.3. Effect of Cd and/or AG on the Expression Level of PM H^+^-ATPase Genes

To evaluate the expression level of *CsHA* genes in the roots of seedlings treated with cadmium and/or AG, a real-time PCR assay was performed. We noticed that the relative expression of PM H^+^-ATPase genes in cucumber roots was affected by Cd. Increased *CsHA2*, *CsHA4*, and *CsHA8* transcript levels were observed. On the other hand, the expression of other genes (*CsHA1*, *CsHA3*, *CsHA9*, and *CsHA10*) remained unchanged (Figure 4). Additionally, it was shown that the increase in the expression level of three isoforms (*CsHA2*, *CsHA4*, *CsHA8*) was completely diminished when the plants were treated with cadmium and then with AG (Figure 4). It is worth mentioning that the transcript level of genes encoding plasma membrane H^+^-ATPase in the control cucumber roots differed significantly (Table 2). The smallest gene expression was found in the case of the isoforms *CsHA1* and *CsHA10*. In contrast, the greatest levels of gene expression were demonstrated for isoforms *CsHA2*, *CsHA3*, and *CsHA8*.

### 2.4. NO Level in Roots of Cucumber Seedlings Treated with Cd and/or AG

Polyamine catabolism may produce nitric oxide. It was found that the endogenous level of NO increased about 130% in the roots of plants exposed to 10 µM Cd for three days in comparison to the control (Figure 5). However, when the cadmium-stressed plants were transferred to the medium containing the DAO inhibitor (AG), the NO level significantly decreased in the roots, almost to the value observed in the control roots. This result suggests that the observed increase in the NO level in the tissues treated with Cd was dependent on DAO activity.

### 2.5. DAO-Induced H_2_O_2_ Level under Cadmium Stress

DAO can oxidize Put, leading to H_2_O_2_ accumulation. To explain the role of DAO in the Cd-induced increase of endogenous H_2_O_2_ content, the level of hydrogen peroxide was determined in the roots in the presence of aminoguanidine. Cucumber plants were grown with the addition of 10 µM CdCl_2_ and after 3 days, returned to the control medium without or with AG for another 3 days. The stimulating effect of Cd on H_2_O_2_ accumulation was significantly reduced (Figure 6).

### 2.6. H_2_O_2_-Dependent NO Generation in the Roots of Cucumber Seedlings

To verify the possible role of hydrogen peroxide in the production of NO in cucumber root tissue, plants were treated with 1 mM H_2_O_2_ for 1, 2, and 24 h and the level of NO was analyzed. It was indicated that hydrogen peroxide contributed to the increase in NO levels in plant roots treated with it for 2 h (Figure 7). However, in the roots of plants exposed to H_2_O_2_ for a longer time (24 h), the level of NO was similar as in the control plants not treated with H_2_O_2_.

## 3. Materials and Methods

### 3.1. Plant Material and Chemical Treatments

Cucumber seeds (*Cucumis sativus* L. var. Wisconsin), germinated for 48 h at 25 °C, were transferred to a nutrient medium for 6 days. The plants were grown in the nutrient solution contained: 1 mM MgSO_4_, 5 mM Ca(NO_3_)_2,_ 5 mM KNO_3_, 1 mM KH_2_PO_4_, and microelements: 75 μM ferric citrate, 10 μM MnSO_4,_ 5 μM H_3_BO_4,_ 1 μM CuSO_4_, 0.01 μM ZnSO_4_, and 0.05 μM Na_2_MoO_4_. The plants were cultivated hydroponically without (control) or with 10 µM CdCl_2_ added to the nutrient solutions (pH 5.5) with a 16-h photoperiod (180 μmol m^−2^ s^−1^) at 25 °C during the day and 22 °C during the night. After 3 days, both the control and Cd-treated seedlings (Cd) were transferred to the fresh control nutrient medium (pH 6.5) for the next 3 days. Additionally, some of the control and cadmium treated plants were supplemented with 0.1 mM aminoguanidyne (AG), a DAO inhibitor, which was introduced to the medium after 3 days of cultivation when plants returned to the control medium. Such treatment conditions are compatible with our previous work [18].

### 3.2. Assay of DAO and PAO

Cucumber roots were used as a source of enzymes. DAO and PAO activities were estimated spectrophotometrically by a method based on the colorimetric assay of Δ-pyrroline using putrescine for DAO and spermidine for PAO as substrates [32]. One gram of fresh roots was homogenized at 4 °C in 100 mM K-phosphate (pH 7.0), containing 10 mM dithiothreitol, 10 µM pyridoxal, 0.1% Triton X-100. Homogenate was held 20 min on ice and centrifuged at 15,000× *g* for 20 min at 4 °C. The supernatant was used to determine the activity of diamine oxidase and polyamine oxidase.

The principle of the method is that Δ-pyrroline formed by the enzymatic oxidation of putrescine or spermidine can react with 2-aminobenzaldehyde to produce a yellowish-colored dihydroquinazolinium derivative. A spectrophotometric assay for the determination of Δ-pyrroline was performed according to Holmsted et al. [32], as modified by Federico and Angelini [33]. The 1 mL reaction mixture contained: supernatant, 50 mM K-phosphate, pH 7.5 or 6.5 (for DAO and PAO, respectively), 10 mM putrescine or spermidine (for DAO and PAO, respectively), 50 U catalase, and 0.1% 2-aminobenzaldehyde. The samples were incubated for two hours at 37 °C. After this time, the reaction was stopped by the addition of 125 μL of 10% TCA and centrifuged for 10 min at 10,000× *g*. The absorbance was measured at 430 nm (ε = 1.86 × 10^−3^ mol × cm^−1^).

### 3.3. Plasma Membrane Isolation and PM H^+^-ATPase Activity Determination

To obtain preparations containing highly purified plasma membrane (PM) vesicles, the method described by Kłobus [34] was used. PM was isolated from cucumber roots by an aqueous two-phase system. The PM obtained by this procedure was mainly composed of right-side-out vesicles and was used to determine hydrolytic ATPase activity. Some of the vesicles were turned to the inside-out oriented form using Brij58 and used for the measurements of ATP-dependent H^+^ transport across the PM.

The hydrolytic activity of the PM H^+^-ATPase was determined by measuring the phosphate released from ATP according to the procedure of Gallagher and Leonard [35]. Proton transport activity was determined by measuring the absorbance change of acridine orange at 495 nm (A_495_), according to Kłobus and Buczek [36]. After the equilibration of membranes with reaction medium (at least for 5 min), vesicle acidification was initiated by the addition of 3 mM Mg-ATP. For each combination, passive proton movement across the PM was measured without ATP in the reaction medium.

### 3.4. Determination of Endogenous NO

Nitric oxide concentration in roots was detected by fluorescent microscopy using the fluorescent NO indicator dye DAF-2DA (5,6-diaminofluorescein diacetate). Roots were briefly excised from the plants and incubated for 10 min in the dark in 20 mM HEPES-KOH, pH 7.4, containing 10 µM DAF-2DA. To remove excess fluorophore from the surface, roots were washed for 15 min in fresh HEPES-KOH buffer. NO-associated fluorescence was detected with a Zeiss Axio Image M2 fluorescent microscope using unchanged parameters for every measurement. For fluorescence observation, a Tag-YFP filter with an emission of 524 nm was used. The intensity of green fluorescence in the images was analyzed using Adobe Photoshop CC software and was expressed as the average number of pixels in a green channel on a scale ranging from 0 to 255.

### 3.5. Histochemical Detection of H_2_O_2_ in Cucumber Roots

H_2_O_2_ was identified by immersing whole six-day-old cucumber plants in a 1 mg/mL solution of DAB, in 0.33 mM MES-NaOH, pH 5.5, for 8 h in dark tubes according to the procedure of Thordal–Christensen et al. [37].

### 3.6. Protein Determination

Protein content was measured by the method of Bradford [38] in the presence of 0.02% Triton X-100, using BSA as the standard.

### 3.7. RNA Isolation and Analysis of Transcript Levels

To assess the expression of the genes encoding PM-H^+^-ATPase, *CsHA1* (GenBank accession no. JK693835), *CsHA2* (GenBank accession no. EU735752), *CsHA3* (GenBank accession no. EF375892), *CsHA4* (GenBank accession no. HO054960), *CsHA8* (GenBank accession no. HO054964), *CsHA9* (GenBank accession no. HO054965), and *CsHA10* (GenBank accession no. HO054966), a real-time polymerase chain reaction (PCR) was performed using the LightCycler (2.0) system from Roche Diagnostics. For the standardization of expression of each *CsHA* gene, a gene encoding TIP41-like protein (GenBank accession no. GW881871) was used as the control. Total RNA was isolated from the roots of cucumber seedlings. A total of 50 mg of tissue was ground in a porcelain mortar under liquid nitrogen with Tri Reagent, following the manufacturer’s instructions (Sigma, St. Louis, MO, USA) and then reverse-transcribed into first-strand cDNA with the High-Capacity cDNA Reverse Transcription Kit (Applied Biosystems, Waltham, MA, USA). The cDNA was then used as the template for PCR amplification with the Real-Time 2 × PCR Master Mix SYBR (A&A Biotechnology, Gdańsk, Poland) kit. The following amplification conditions were applied: 30 s at 95 °C, 45 cycles of 10 s at 95 °C, 10 s at 58 °C, and 12 s at 72 °C, with a final melting for 15 s at 65 °C.

### 3.8. Statistical Analysis

The quantitative PCR data was analyzed by the ΔΔCT method using Light Cycler 4.1 software (Roche). At least three independent experiments concerning protein and RNA extractions, the measurements of enzyme activity and gene expression were made, run in triplicate, and the means ± standard deviation (SD) of these results are presented in the figures. Results were statistically analyzed by the Student’s *t*-test. Asterisks in the figures indicate significant differences, (*p* < 0.05 one asterisk, *p* < 0.01 two asterisks, *p* < 0.001 three asterisks), and a statistical tool in R package (R Core Team 2021) was used.

The jamovi project (2021). jamovi. (Version 2.0) [Computer Software]. Retrieved from https://www.jamovi.org.

R Core Team (2021). R: A Language and environment for statistical computing (Version 4.0) [Computer software]. Retrieved from https://cran.r-project.org (R packages retrieved from MRAN snapshot 1 April 2021).

## 4. Discussion

In the presented research, we decided to analyze the responses of plants to stress factors, in which PM H^+^-ATPase plays a key role, using cucumber seedlings as research material. Cucumber (*Cucumis sativus* L.) is a crop plant and an economically important vegetable. According to Statista, in 2020, global production of cucumber increased to over 91 million metric tons, making cucumber the third most-produced vegetable. The nutritional value of the cucumber is low, but its delicate flavor makes it popular for salads and relishes. Because of the small number of genes, rich diversity of sex expression, suitability for vascular biology studies, and short lifecycle cucumber is being developed as a new model plant [39].

Polyamines are molecules, which not only participate in plant growth and development, but are also involved in the adaptation of plants to environmental stresses. Polyamine catabolism seems to be particularly important in the process of plant adaptation to unfavorable environmental conditions. Two kinds of enzymes are involved in PA catabolism, copper-dependent diamine oxidase, and flavin adenine dinucleotide (FAD)-dependent polyamine oxidase. Both DAO and PAO are responsible for the oxidation of PAs, which occurs together with the production of H_2_O_2_ [23]. Analyses of the activity of both enzymes in cucumber seedlings indicated that PAO activity was very low whereas DAO activity was almost sixteen times greater (Figure 1). Our results are in line with the data provided by Moschou et al. [25]. In plants, DAO occurs at high level in dicots, particularly pea, chickpea, lentil, and soybean seedlings, loosely associated with cell wall. In contrast, PAO is highly expressed in monocots. Diamine oxidase is also often found in rapidly growing tissues [24]. We have also shown that the treatment of cucumber plants with cadmium changed the activity of the DAO but had no effect on the activity of PAO in roots. Under all of the tested conditions, PAO activity maintained at a very low level, regardless of the treatment (Personal communication, M. Janicka, available in repository). Considering the available data [25] and the results obtained as well as working with dicot and very young plants (*Cucumis sativus* seedlings), showing the very low level of PAO activity, in further experiments, only the activity of DAO was analyzed. Treatment of plants with cadmium stimulated DAO activity twice (Figure 2). AG (a DAO inhibitor) significantly reduced enzyme activity in both the control and cadmium-treated plants (Table 1). The involvement of amine oxidases in polyamine catabolism is related to their role in plant defense responses. Under aluminum stress, an increase of diamine oxidase activity in pea root nodules was observed. This increase in DAO activity was responsible for the enhanced accumulation of hydrogen peroxide in nodules [28].

In our experiments, treatment of the cucumber seedlings with cadmium increased the H^+^-ATPase activity in the plasma membrane fraction (Figure 3A,B). This enzyme functions as a proton pump, which couples ATP hydrolysis to proton transport and creates an electrochemical H^+^ gradient across plasma membrane. This gradient is used by secondary transporters [40]. Aside from the regulation of physiological processes, the plasma membrane proton pump participates in the adaptation of plants to stress factors [13]. Heavy metals often lead to disturbances in plants including membrane damage and ion imbalance. In such conditions, maintaining ionic homeostasis and replacing the loss of substances in reparative processes is a significant matter. Because the active transport of ions across the plasma membrane requires increased generation of a proton gradient by PM H^+^-ATPase, regulation of its activity is an important cellular mechanism for stress tolerance. To investigate whether the increased enzyme activity observed in plants treated with Cd could be attributed to DAO action, the effect of AG, the inhibitor of DAO, on the membrane proton pump was studied in cucumber seedlings (Figure 3A,B). The stimulating effect of Cd on enzyme activity was totally abolished by AG, suggesting that DAO may be involved in the modification of PM proton pump functioning under cadmium stress. Since the AG does not change the activity of the plasma membrane proton pump under control conditions (Figure 3), it seems that the participation of DAO in the activation of the enzyme (PM H^+^-ATPase) is important only when the plants are exposed to cadmium stress.

Polyamine catabolism may be involved in the regulation of gene expression of proteins associated with other processes. In the *Arabidopsis* mutant defective in *AtPAO4* (polyamine oxidase gene), altered expression of genes related to abiotic stress responses and the metabolism of flavonoids and lignins was observed [24]. Hydrogen peroxide formed in the apoplast during the catabolism of PAs triggers a cascade of reactions, leading to increased expression of specific genes encoding superoxide dismutase, ascorbate peroxidase, pathogenesis-related proteins, kinases, transcription factors, and other stress response proteins. The application of spermine to tobacco leaves, mimicking apoplastic accumulation of polyamines as a result of incompatible plant–pathogen interactions, increased the expression of marker genes for hypersensitivity reactions. On the other hand, the observed stimulation of gene expression was decreased by inhibitors of diamine and polyamine oxidase. This indicates the participation of H_2_O_2_, resulting from the polyamine decomposition, in the regulation of the expression of these genes [24]. In our experiments, cadmium-induced changes in PM H^+^-ATPase activity were observed and these changes were suggested to be due to the action of DAO (Figure 3A,B). The modification of the activity of plasma membrane proton pump can take place at the level of gene transcription. For this reason, the expression level of *CsHA* genes (encoding isoforms of plasma membrane H^+^-ATPase in cucumber) was analyzed in seedling roots treated with cadmium and/or AG. We noticed that relative expression of PM H^+^-ATPase genes in cucumber roots was affected by Cd. An increase in the *CsHA2*, *CsHA4*, and *CsHA8* transcript levels was observed (Figure 4). Moreover, this increase was completely diminished by the presence of AG. It can therefore be suggested that the hydrogen peroxide generated during the breakdown of polyamines may be responsible for both changes in the expression of genes encoding PM H^+^-ATPase and an increase in its activity at the protein level. Earlier, we observed that treatment of cucumber seedlings with H_2_O_2_ contributes to increased expression of PM H^+^-ATPase genes in roots [41].

We have shown that DAO activity generates H_2_O_2_ in cucumber roots under the cadmium stress condition (Figure 6). DAO can oxidize Put to induce H_2_O_2_ accumulation [23]. Our previous studies have found that in cucumber plants treated with cadmium, accumulation of hydrogen peroxide in roots was observed [42]. To examine the role of DAO in the Cd-induced increasing of H_2_O_2_ content, aminoguanidine was used. It was confirmed that Cd enhances H_2_O_2_ accumulation via the activity of DAO (Figure 6). Previous studies have shown that the plasma membrane proton pump plays an essential role in the adaptation of cucumber seedlings to cadmium stress [17,18,42]. New data suggest that diamine oxidase, through the production of hydrogen peroxide, may be an important element of the signal transduction pathway, leading to the modification of PM H^+^-ATPase activity under cadmium stress conditions.

Nitric oxide appears to be another player in plant reactions to cadmium. We have observed that short time (2 h) treatment of plants with H_2_O_2_ promotes the formation of NO in the roots of cucumber seedlings (Figure 7). Both nitric oxide and hydrogen peroxide are involved in plant responses to biotic as well as abiotic stresses [15,43,44]. The review of Gill et al. [31] clearly indicated the importance of nitric oxide in cadmium stress tolerance in crop plants. We have demonstrated earlier that pretreatment of cucumber plants with the NO donor (sodium nitroprusside) or with H_2_O_2_ increased the hydrolytic and transporting activities of PM H^+^-ATPase [15]. In this study, it was observed that hydrogen peroxide could contribute to an increase in endogenous NO levels, suggesting that the stimulation of H^+^-ATPase in the plasma membrane under Cd stress could be caused by the action of DAO, which, through the production of H_2_O_2_, could lead to an increase in NO level. In plants, the signal transduction pathways are still elucidated. In a number of abiotic and biotic responses, H_2_O_2_ generation occurs in parallel with NO, and these molecules can act both synergistically and independently [45,46]. Tun et al. [29] reported that PAs induced NO biosynthesis in *Arabidopsis thaliana* seedlings. Additionally, in *Arabidopsis,* copper amine oxidase 1 (CuAO1) contributes to ABA and PA-induced NO biosynthesis [47]. Moreover, copper amine oxidase 8 regulates arginine-dependent NO production in *Arabidopsis* [30]. NO biosynthesis as a result of PA catabolism, catalyzed by DAO, may explain many functions of PA mediated stress responses. In our study, the endogenous levels of NO increased in plants treated with Cd (Figure 5). However, when the cadmium-stressed plants were additionally treated with the DAO inhibitor, the NO level decreased. This result suggests that the increase in NO content in the tissues treated with Cd is dependent on DAO activity.

## 5. Conclusions

In summary, this work characterizes for the first time the role of DAO in the adaptation of plants to cadmium stress via regulation of PM H^+^-ATPase activity. Treatment of cucumber plants with Cd stimulated both DAO and PM H^+^-ATPase activities in cucumber seedling roots. The stimulating effect of Cd on the PM H^+^-ATPase was reduced by AG. Moreover, it was observed that H_2_O_2_ produced by DAO promotes the formation of NO in the roots of plants. It seems that DAO may be an element of the signal transduction pathway, leading to enhanced PM H^+^-ATPase activity under cadmium stress. Our results provide new insights into the PA mediated signaling, in which proton pump may function as a potential target, and both H_2_O_2_ and NO act as mediators.

## Figures and Tables

**Figure 1 ijms-24-00262-f001:**
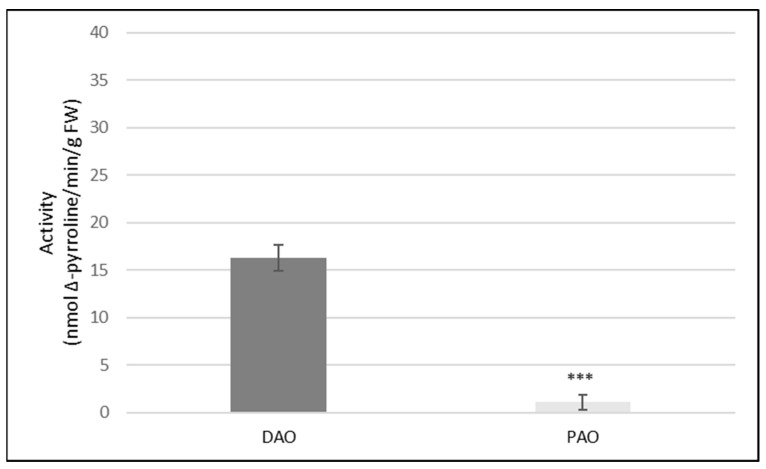
Activities of DAO and PAO in the roots of cucumber seedlings. Amine oxidase activities were measured in control cucumber plants, according to the method described in the Materials and Methods. Results are the means ± SD of five independent experiments with each experiment performed in six replicates. There was a significant difference between DAO and PAO activity (asterisks indicate the significant differences, where *** *p* < 0.001).

**Figure 2 ijms-24-00262-f002:**
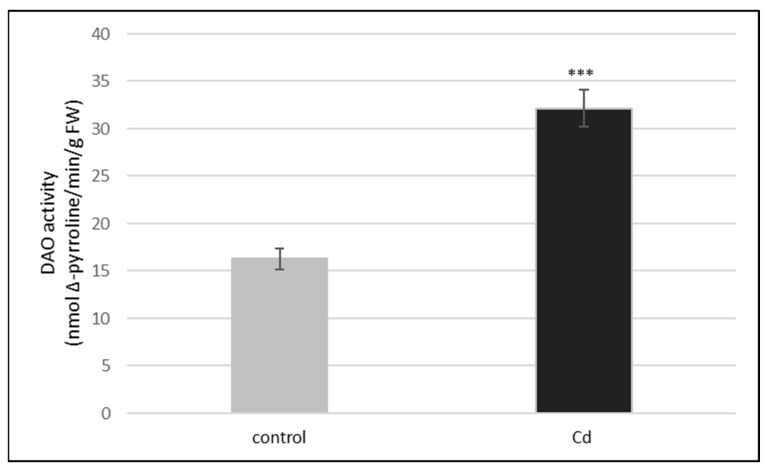
DAO activity in the roots of cucumber seedlings treated with 10 µM Cd. DAO activity was measured in the control plants and plants, which after 3 days of treatment with Cd were transferred to the control conditions for another 3 days (Cd). The data are presented as means ± SD from three independent experiments with each experiment performed in six replicates. There was a significant difference between DAO activity in the control and Cd treated plants (asterisks indicate the significant differences, where *** *p* < 0.001).

**Figure 3 ijms-24-00262-f003:**
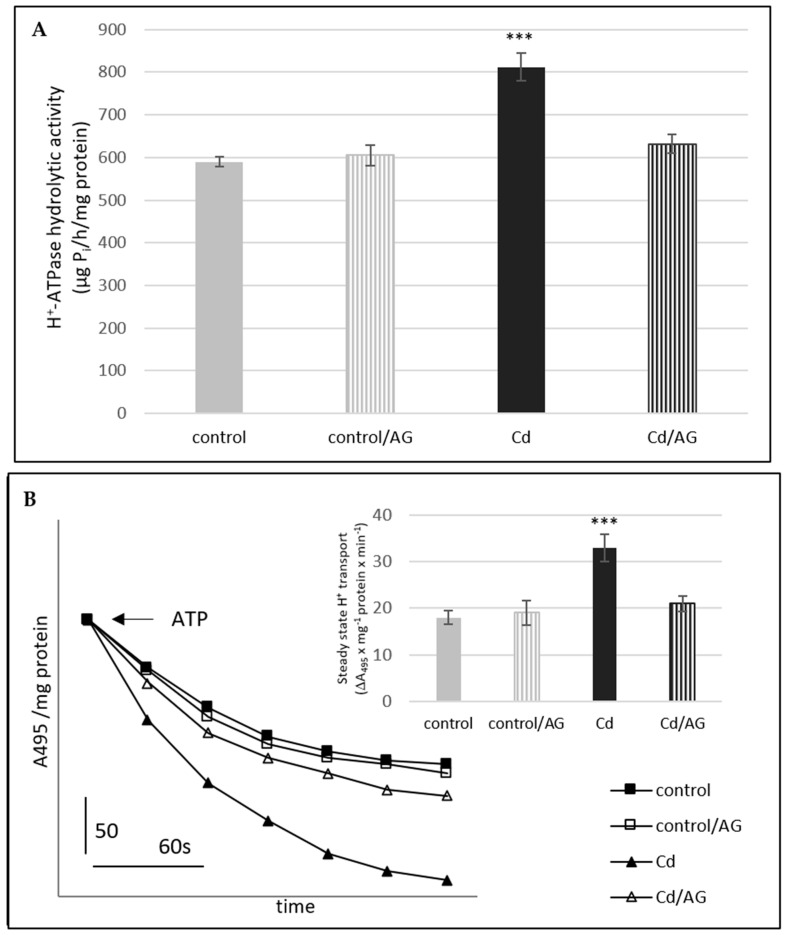
Effect of cadmium and/or AG on the hydrolytic activity of H^+^-ATPase (**A**) and the ATP-dependent proton transport (**B**) measured in the plasma membrane vesicles. The plasma membranes (50 μg of protein) were isolated from control roots (control), roots of plants treated for 3 days before harvesting with aminoguanidyne added to control nutrient solution (control/AG), roots of plants, which after 3 days of treatment with 10 µM Cd were transferred to the control conditions for another 3 days (Cd), and the roots of plants, which after 3 days of treatment with 10 µM Cd were transferred to the control nutrient solutions with aminoguanidyne (Cd/AG). Hydrolytic activity of H^+^-ATPase was measured as described in the Materials and Methods. Results are the means ± SD of three independent experiments with each experiment performed in triplicate (**A**). After equilibration of membranes with the reaction medium (for at least 5 min), vesicle acidification was initiated by the addition of ATP to give a final concentration of 3 mM. The formation of a ΔpH gradient in the vesicles was monitored as the changes in acridine orange absorbance (A_495_). The values presented (**B**) are representative for the results obtained in three independent experiments with each experiment conducted in triplicate. The results in the inner diagrams (the steady state of H^+^ transport are the means ± SD from those three independent experiments (asterisks indicate the significant differences, where *** *p* < 0.001).

**Figure 4 ijms-24-00262-f004:**
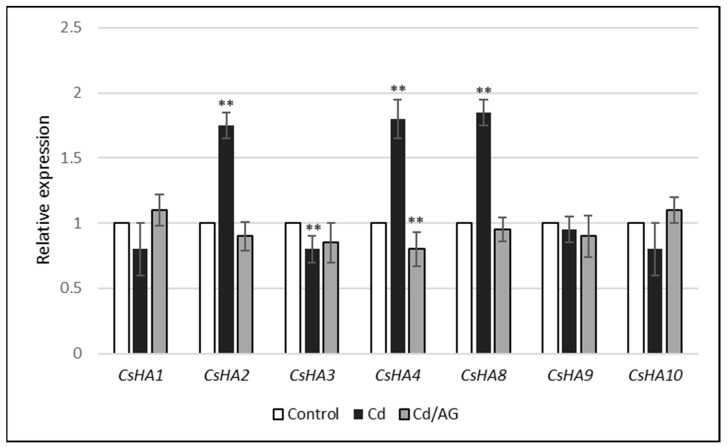
Relative expression of PM H^+^-ATPase genes in cucumber roots exposed to Cd and/or AG. To determine the expression of PM H^+^-ATPase genes, real-time PCR analysis was performed as described in the Materials and Methods. The RNA was isolated from the control roots (control) and roots treated with 10 µM Cd (Cd) or 10 µM Cd and AG (Cd/AG). Values are the means of three replications. Error bars represent SD (asterisks indicate the significant differences, where ** *p* < 0.01).

**Figure 5 ijms-24-00262-f005:**
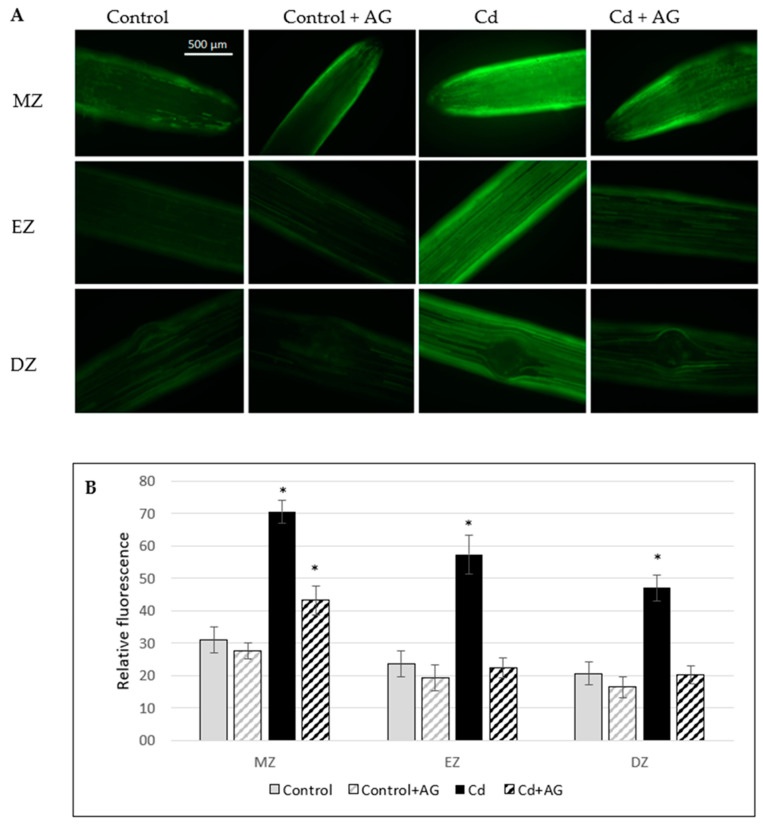
Bio-imaging of NO level in the roots of cucumber seedlings treated with Cd in the presence or absence of aminoguanidyne (AG) (**A**) and the mean related fluorescence density ± SD (**B**). After 3 days of treatment with Cd, plants were transferred to the control conditions for another 3 days (Cd) or to the control nutrient solutions with aminoguanidyne (Cd + AG). At the same time, control plants were also treated with aminoguanidyne (control + AG). NO production was monitored by labeling with the NO-specific fluorescent dye DAF-2D and imaged using fluorescent microscopy. The images are representative for at least three roots for each treatment from three independent replications of the experiment. Asterisks on the figure indicate significant differences (* *p* < 0.05). The intensity of green fluorescence in the images was analyzed and expressed as the average number of pixels in green channel on a scale ranging from 0 to 255. MZ—meristematic zone, EZ—elongation zone, DZ—differentiation zone.

**Figure 6 ijms-24-00262-f006:**
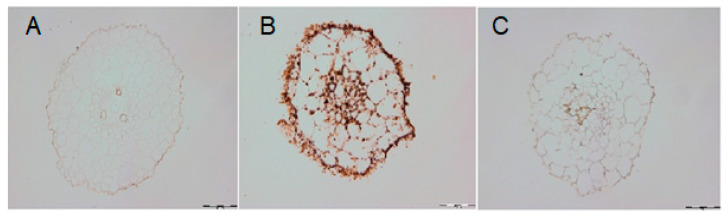
DAB staining and visualization of hydrogen peroxide in the roots of cucumber seedlings: control plants (**A**), plants treated with Cd (the plants after 3 days of treatment with Cd were transferred to the control conditions for another 3 days) (**B**) and plants, in which the heavy metal was withdrawn after 3 days, and then aminoguanidyne was added for the next 3 days (**C**).

**Figure 7 ijms-24-00262-f007:**
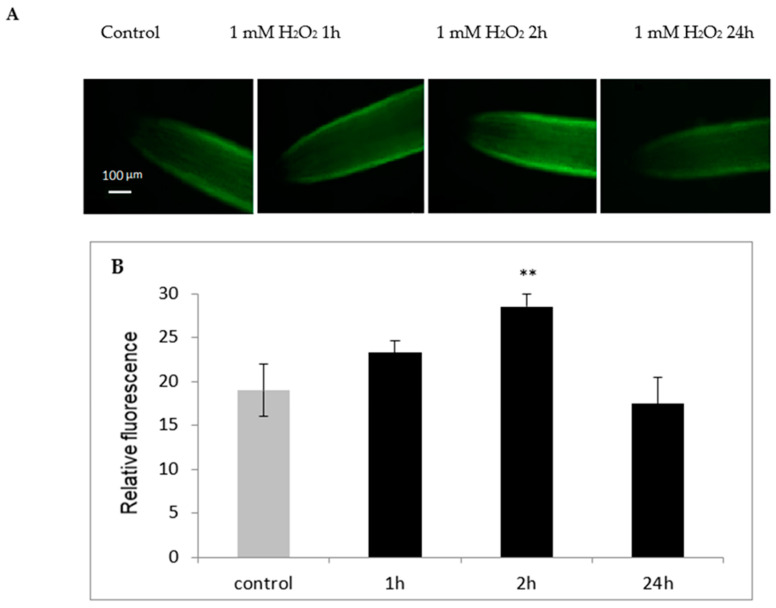
Bio-imaging of NO level in the roots of cucumber seedlings treated with H_2_O_2_ for 1, 2, and 24 h (**A**) and the mean related fluorescence density ± SD (**B**). Control plants were not treated with H_2_O_2_. NO production was monitored by labeling with the NO-specific fluorescent dye DAF-2D and imaged using fluorescent microscopy. The images are representative for at least three roots for each treatment from four independent replications of the experiment. Asterisks on the figure indicate significant differences (** *p* < 0.01).

**Table 1 ijms-24-00262-t001:** Inhibition of DAO activity by AG. DAO activity was measured in the control plants (control, −AG) or in plants treated for 3 days with aminoguanidyne added to the nutrient solution before the collection (control, +AG) as well as in plants, which after 3 days of treatment with Cd were transferred to the control conditions without (Cd, −AG) or with aminoguanidyne (Cd, +AG) for another 3 days.

	DAO Activity (pmol Δ-Pyrroline/min/g FW)	
	−AG	+AG
Control	16.3 (±1.1)	2.7 (±0.7)
Cd	32.1 (±1.9)	3.3 (±1.2)

**Table 2 ijms-24-00262-t002:** Transcript level of genes encoding plasma membrane H^+^-ATPase in the roots of *Cucumis sativus* L. For the normalization of expression of each *CsHA* gene, a gene encoding TIP41-like protein was used as the internal standard.

Gene	Transcript Level (Fluorescence Units)
*CsHA1*	0.01
*CsHA2*	14.23
*CsHA3*	13.25
*CsHA4*	0.83
*CsHA8*	7.33
*CsHA9*	2.17
*CsHA10*	0.18

## Data Availability

The data presented are available in this manuscript and in the repository at: https://www.repozytorium.uni.wroc.pl/dlibra/publication/139358/edition/128721#, https://www.repozytorium.uni.wroc.pl/dlibra/publication/139336/edition/128680 (accessed on 10 September 2021).

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
