# Peer review of "Involvement of Diamine Oxidase in Modification of Plasma Membrane Proton Pump Activity in Cucumis sativus L. Seedlings under Cadmium Stress"

_ijms, 2022, doi:10.3390/ijms24010262_

Round 1

Reviewer 1 Report

This is an interesting paper, which shows the involvement of diamine oxidase (DAO) in stimulation of the proton pump, when important food crop, cucumber, is exposed to Cd stress. H2O2 and NO are also involved in the signaling sequence. The greater activity of the proton pump is needed to power the Cd/H+ antiporter, which removes Cd from the cytoplasm.

The paper is well written, but the “introduction” would read better with more paragraph differentiation.

The figures are clear, except in Fig. 4 and particularly Fig. 5 the sample patterns are too small to be distinguished.

Author Response

Thank you very much for revision of  our manuscript. We followed carefully the comments and, according to them, improved our paper.

1) “The paper is well written, but the “introduction” would read better with more paragraph differentiation.” We agree with the comment and modified the layout of the introduction, dividing it into appropriate fragments.

2) “The figures are clear, except in Fig. 4 and particularly Fig. 5 the sample patterns are too small to be distinguished.” All figures in the manuscript have been corrected for easy readability.

Reviewer 2 Report

Please revise the keywords to enhance the manuscript's reach and try to avoid or limit the words from the title of the manuscript.

In line 48: contribute can be replaced with mediate or induce; please check.

The introduction is unnecessarily elaborated; especially the source of Cd; these are well-known facts so can be briefed in 1-2 sentences.

In the whole manuscript, there is extensive use of personal pronouns, so please avoid or minimise them.

In cucumber seeds, darkness has a role in the promotion of seed germination; please explain.

In the manuscript, sometimes, commas have been used in place of decimals, pay attention.

In Figure 1, a table with results has been provided but not in others; any specific reason; please explain.

The quality of the figures must be improved.

In Figure 4, there is no SD bar for the control group; why? There was no variation in the control replications; please explain.

Please pay attention to the use of acronyms throughout the manuscript.

Please revise the conclusion of the manuscript.

Author Response

First of all we would like to thank you for detailed revision of  our manuscript.  We followed carefully the comments and improved our paper, hoping that all those changes make our manuscript much more convenient.

1) „Please revise the keywords to enhance the manuscript's reach and try to avoid or limit the words from the title of the manuscript.” Thank you for your attention, we have removed words that were repeated in the title and added new one keywords.

 2) „In line 48: contribute can be replaced with mediate or induce; please check.” We changed the word conribute to mediate.

3) "The introduction is unnecessarily elaborated; especially the source of Cd; these are well-known facts so can be briefed in 1-2 sentences.” We agree with this comment and removed redundant information about cadmium from the introduction.

4) "In the whole manuscript, there is extensive use of personal pronouns, so please avoid or minimise them.” We have minimized the use of personal pronouns.

5) „In cucumber seeds, darkness has a role in the promotion of seed germination; please explain.” Darkness had no effect on cucumber seed germination. The temperature in the thermostat was important, there was no access to light, and that's why we wrote about germination in the dark. We have removed this information (word) from the manuscript.

6) „In the manuscript, sometimes, commas have been used in place of decimals, pay attention.” Thank you, we have changed it.

7) „In Figure 1, a table with results has been provided but not in others; any specific reason; please explain.” The arrangement of Table 1 and Figure 1 was unfortunate. It gave the impression that the data from the Figure and the Table were the same. However, the table and figure show different results. We have changed the position of Table 1 and Figure 1 so that it does not cause confusion.

8) „The quality of the figures must be improved.” The quality of figures has been improved.

9) „In Figure 4, there is no SD bar for the control group; why? There was no variation in the control replications; please explain.” Quantitative PCR data were analyzed by ΔΔCT using Light Cycler 4.1 software. The results were normalized, which means that the control was always 1, the measurement error was calculated in relation to this normalized result.

10) „Please pay attention to the use of acronyms throughout the manuscript.” Thank you, we try pay attention to the use of acronyms throughout the manuscript.

11) „Please revise the conclusion of the manuscript.” Thank you very much for this comment, our conclusion was not appropriate. We changed the conclusion of the manuscript.

Once again thank you very much for review.

Sincerely

Małgorzata Janicka

Round 2

Reviewer 2 Report

In the revised manuscript, the authors have addressed all the comments; so after careful consideration, I'm recommending it for acceptance.